

# The Atlantic Ocean's Decadal Variability in mid-Holocene Simulations using Shannon's Entropy

Iuri Gorenstein[1], Ilana Wainer[1], Francesco S. R. Pausata[2], Luciana F. Prado[3], Pedro L. Silva Dias[4], Allegra N. LeGrande[5], Clay R. Tabor[6], and William R. Peltier[7]

[1]Departamento de Oceanografia Física, São Paulo, SP, Brazil
[2]Centre ESCER (Etude et la Simulation du Climat à l'Echelle Regionale) and GEOTOP (Research Center on the Dynamics of the Earth System), Department of Earth and Atmospheric Sciences, University of Quebec, Montreal, Montreal, QC, Canada
[3]Faculdade de Oceanografia, Universidade do Estado do Rio de Janeiro, Rio de Janeiro, Brazil
[4]Instituto de Astronomia, Geofísica e Ciências Atmosféricas, Universidade de São Paulo, Departamento de Ciências Atmosféricas, São Paulo, Brazil
[5]NASA Goddard Institute for Space Studies, and Center for Climate Systems Research, Columbia University, New York, USA
[6]Department of Geosciences, University of Connecticut, USA
[7]Department of Physics, University of Toronto, Canada

**Correspondence:** Iuri Gorenstein (iuri.gorenstein@usp.br)

**Abstract.** Accurate simulation of mean climate and variability is crucial for numerical climate models. Traditional methods assess variability using two-dimensional standard deviation fields, like sea surface temperature (SST) and precipitation, to identify key regions. However, this approach can overlook large-scale patterns, such as ocean modes of variability, used in traditional climatology and oceanography to define climate variability. We propose a method incorporating large-scale climate patterns to evaluate and compare decadal variability in four coupled models (EC-Earth, GISS, iCESM, and CCSM-Toronto). Shannon's Entropy compares the models' sensitivity to different scenarios: pre-industrial period, mid-Holocene with default vegetation, and mid-Holocene with prescribed Green Sahara conditions. Results show contrasting model responses, with little consensus on the effects of Green Sahara vegetation and orbital forcing. Three models (EC-Earth, iCESM, and CCSM-Toronto) show reduced precipitation variability under Green Sahara conditions, but with differing SST responses. The GISS model shows minimal effects on variability. Additionally, reducing dust in the Green Sahara scenario significantly impacted EC-Earth's model, increasing precipitation while decreasing SST variability. These findings highlight the diverse representations of climate variability across models and offer a new methodology for comprehensive model analysis.

## 1 Introduction

A system's variability can be perceived as the absence of uniformity across multi-scales (Sang, 2013). Earth's climate can be interpreted as a high dimensional chaotic system (highly dependant on initial conditions), with many feedbacks and interactions among an extensive set of particles and radiation (Ghil and Lucarini, 2020; Kwiecien et al., 2022). Since this system is in constant motion, climate internal variability denotes how we perceive the climate system driven by its intrinsic dynamics, regardless of any transient external forcings (Flato et al., 2014). Numerical climate models are used to extrapolate our limited



observational data set and study not only the climate's internal variability, but its changes, and debate past and future scenarios
of Earth's climate.

The study of interannual precipitation patterns in the tropical and South Atlantic regions (15N - 30S, 60W-20E, region defined in the maps from Figs. 1, 2 'a'), as well as the adjacent continents, provides critical insight into the dynamics of the Intertropical Convergence Zone, one of the most distinctive characteristics of Earth's climate variability. This major zone of convection and surface wind convergence is significantly influenced by variations in wind, sea level pressure, and sea
surface temperature (SST) gradients, which can displace the zone both within and across hemispheres (Dhrubajyoti et al., 2019; Hounsou-Gbo et al., 2019; Atwood et al., 2020). Such variations are key not only on an annual scale but also have important implications for decadal precipitation changes, further modulated by the ocean's modes of variability, such as the Atlantic Equatorial Mode (AEM), Atlantic Meridional Mode (AMM) and South Atlantic Subtropical Dipole (Gorenstein et al., 2023). Oscillating at lower frequencies, these modes play an essential role in the displacement of convergence zones (Deser
et al., 2010). The interaction between ocean and atmospheric dynamics, rooted in non-linear processes within large-scale climate phenomena, establishes a robust coupling between precipitation and SST on a decadal scale. This complex relationship underscores the difficulty in predicting precipitation and SST anomalies due to their reliance on a wide range of interacting climate variables (Deser et al., 2012).

Studying past climates is a practical way to apply numerical climate models to study diverse equilibrium states of Earth's
climate. In the mid-Holocene (MH) period, approximately 5000-7000 years Before Present, we observe a climate characterized by differential summer insolation across hemispheres, leading to the "Holocene Thermal Maximum" (Berger, 1988; Liu et al., 2002; Bova et al., 2021). Despite its potential as an analog for future climate scenarios (Burkea et al., 2018; Kaufman et al., 2020), the warming during this period was neither uniform across the globe nor consistent throughout the year, exhibiting notable disparities across hemispheres and seasons (Berger, 1978; Zhao and Harrison, 2012; Huo et al., 2021). This period
saw an intensification of monsoon systems in the Northern Hemisphere, while precipitation in the Southern Hemisphere was reduced compared to the present day (Liu et al., 2004; Wanner et al., 2008; Smith and Mayle, 2017). Furthermore, other climate forcings and feedbacks played an important role during the MH, such as the expansion of vegetation in areas that are currently deserted in northern Africa (the Green Sahara - GS) and Asia. Such changes had a remarkable impact on global temperature and precipitation patterns and contributed to an intensification of the Atlantic Meridional Overturning Circulation, impacting
vegetation in regions dependent on the South American summer monsoon, such as the southwest Amazon and parts of Brazil (Smith and Mayle, 2017; Gorenstein et al., 2022a).

The geophysical mechanisms behind the Atlantic Ocean modes coupling with pressure and wind driving decadal precipitation anomalies were unraveled using observational data in Gorenstein et al., 2023. When examining this region's dynamics in climate simulations, a more fundamental question arose regarding how to assess its decadal variability. Since numerical
models present biased climate representations to observational data and among themselves (Dhrubajyoti et al., 2019), their climate variability is not usually measured concerning large climate patterns such as ocean modes, instead, it is measured using point-wise standard deviation and frequency spectra (Olonscheck and Notz, 2017; Pendergrass et al., 2017). This motivated




us to create a new method using ocean modes and their precipitation counterparts to quantify decadal variability in numerical climate models.

In this study, we define climate variability with concepts from statistical mechanics, oceanography, and climatology. Applying Shannon's entropy to measure the organization of the Atlantic Ocean modes and their precipitation counterparts, we compare climate variability among different numerical models and scenarios. Simulations from four different Earth System Models (EC-Earth, iCESM, CCSM-Toronto, and GISS) are analyzed to study pre-industrial (PI) climate and mid-Holocene (MH) experiments. The MH experiments include conditions with ($MH_{GS}$) and without ($MH_{PMIP}$) vegetation changes in northern Africa, as well as experiments incorporating parametrized changes in dust reduction, lake extensions, and soil moisture levels (see Methods).

## 2    Material and Methods

Simulations from four numerical climate models (EC-Earth, iCESM, CCSM-Toronto, and GISS) are used to study decadal climate variability of the tropical and South Atlantic Sea Surface Temperature (SST) and precipitation, and its sensibility to prescribed parametrizations in pre-industrial (PI) and mid-Holocene (MH) simulations. Details defining each numerical simulation are described below.

### 2.1    Data

#### *EC-Earth*

The European Consortium Earth System Model Version-3 (EC-Earth) scenarios analyzed in this study were: PI (100 years run B405 and 200 years run B400), $MH_{PMIP}$ (100 years run Z6KA and 200 years run B6KA), $MH_{GS}$ (100 years run G105 and 50 years run G100 ) and $MH_{GSdr}$ with dust reduction (100-year run G506, and 200-year run G501).

EC-Earth standard configuration consists of the atmosphere model IFS including the land surface module HTESSEL and the ocean model NEMO3.6 with the sea ice module LIM3. Coupling variables are communicated between the different component models via the OASIS3-MCT coupler (Döscher et al., 2021). The EC-Earth model is used to contribute to CMIP6 in several configurations, for example, the EC-Earth3-Veg configuration which couples the LPJ-Guess dynamic vegetation model (Smith et al., 2014) to the atmosphere and ocean model; however, the performance of EC-Earth3 and EC-Earth3-Veg is very similar (Wyser et al., 2020).

#### *CESM models*

The Community Earth System Model (CESM) outputs from different scenarios used were from CCSM-Toronto: PI, MH, GS, and GS with soil and lake input (100 years run each); and iCESM: PI, $MH_{PMIP}$, $MH_{GS}$ (100 years run each).

The CESM models used here are from the Cmip6 multi-model ensemble. The CCSM-Toronto simulations are a PMIP experiment for the mid-Holocene with Green Sahara and mid-Holocene with soil and lake inputs made by the University of



Toronto (UofT), Canada. The model configuration was made by UofT-CCSM4 (2014), atmosphere from CAM4 (finite-volume dynamical core; 288 x 192 longitude/latitude; 26 levels; top level 2 hPa) (Peltier and Vettoretti, 2014); ocean: POP2; sea ice:

CICE4; land: CLM4. The iCESM simulations used in this study were first presented in Tabor et al. (2020). iCESM is configured with CAM5, POP2, CLM4, CICE4, and RTM (Brady et al., 2019; Hurrell et al., 2013). The atmosphere and land have a 1.9 × 2.5° horizontal resolution, and the ocean and sea ice have a nominal 1° horizontal resolution. Model configurations include a preindustrial simulation (1850 CE), a mid-Holocene simulation with a 6-ka orbit and greenhouse gases and preindustrial vegetation, and a mid-Holocene Green Sahara simulation with a 6-ka orbit and greenhouse gases and a vegetated Sahara.

Dust emissions from the Sahara are reduced in the mid-Holocene Green Sahara simulation. For additional model configuration details, see Tabor et al. (2020).

### GISS

The scenarios from the Goddard Institute for Space Studies Model E2 coupled with the Russel ocean model (GISS-E2-R) were: PI, $H_{PMIP}$, and $MH_{GS}$ with North African vegetation only, $MH_{GSEX}$ with Extra-Tropical vegetation only, and two

runs of $MH_{GSALL}$ with Full vegetation (100 years run each).

All runs—except for GS Full Vegetation Run 1—use updated aerosol and ozone inputs for non-anthropogenic simulations and apply the Green Sahara vegetation based on Nancy Kiang's regression on leaf area index (Kiang, 2002). In contrast, GS Full Vegetation Run 1 employs a regression script based on the Köppen–Geiger classification to prescribe the leaf area index (Sohoulande, 2023). Several experiments have been set up for the last millennium with GISS due to uncertainties in

past forcings and their effects, with different combinations of solar, volcanic, and land use/vegetation (Colose et al., 2016; LeGrande et al., 2015; Bühler et al., 2022).

Table 1: Data used in this study

| Model | Experiments | Reference |
|-------|-------------|-----------|
| EC-Earth | $PI, MH, MH_{GS}, MH_{GSdr}$ (dust reduction) | Döscher et al. (2021) |
| iCESM | $PI, MH, MH_{GS}$ | Tabor et al. (2020) |
| CCSM-Toronto | $PI, MH, MH_{GS}, MH_{GSsl}$ (soil and lake) | Peltier and Vettoretti (2014) |
| GISS | $PI, MH, MH_{GS}, MH_{GSna}$ (North Africa vegetation) $MS_{GSex}, MH_{GSall}$ (Extra-tropical and Full vegetation) | Schmidt et al. (2014) |





*Low frequency filters*

To study the decadal variability of the Atlantic Ocean, decadal filters were applied to all data sets. These filters are calculated
with the simple decadal mean from the original monthly time series.

## 2.2  Methods

*Principal Component Analysis and the phase space*

The Principal Component (PC) analysis (also known as Empirical Orthogonal Functions - EOF) is a technique used to reduce
data dimensionality. When studying a high-dimension stochastic dynamical system, such as numerical climate model simu-
110 lations, finding relevant statistical information emerging from the physics of the system can be inefficient and overwhelming
(Haykin, 2009). The PC analysis derives a new set of orthogonal coordinates from your data, maximizing the variance from
each component in a decreasing fashion, enabling a drastic dimensionality reduction of your data set while preserving its
variation (Jolliffe, 2002).

There are three main ocean modes of variability controlling the decadal climate variability in the tropical and South Atlantic
region (Deser et al., 2010). Together with pressure gradients and wind anomalies, they induce precipitation in the ocean and
adjacent continents (Gorenstein et al., 2023). We represent the data using the three leading principal components of sea surface
temperature and precipitation in the tropical and South Atlantic. Together, these components explain about 50% of the total
variance across all simulations (21% from the first PC, 17% from the second, and 12% from the third). While PC patterns can
vary between model simulations and may not exactly match observations, we limit our analysis to three components based on
their established physical relevance as drivers of observed decadal variability in the region. This also avoids the complexity of
working in higher-dimensional phase space and ensures our analysis remains grounded in physically meaningful modes.

In this approach, we extracted two distinct PC phase spaces from the dataset: one for SST and another for precipitation. Each
PC phase space consists of the three principal components derived from the combined simulations, encompassing all models
and scenarios. This process results in a unified 3-dimensional phase space for each variable, providing a consistent framework
to analyze and compare the variability across different simulations.

*The trajectories in phase space*

Projecting a simulation series onto the 'n' principal components of the set of simulations generates a trajectory in an n-
dimensional phase space (not shown).This trajectory is determined by the continuous PC indices ('a' in Figs. 1 and 2). Defining
negative, neutral, and positive phases for each index-using a threshold linked to Shannon's Entropy (as described in Section
2.3)-, we discretize this space into $n^3$ states.

Since we choose to represent the tropical and South Atlantic system with its 3 main components, our discrete phase space
is a set of 27 possible states for SST and 27 possible states for precipitation (Figure S1 from Supplementary Material). The
trajectories of each simulation run in phase space can be depicted with directed graphs ('b' in Figs. 1 and 2).



Each node in these graphs represents a state of the system at a particular time step, with the node's size indicating the number

of months the system remained in that state. Nodes that appear darker have a higher degree, meaning they are connected to more transitions to and from other states. This indicates that the system frequently returns to the same state, which results in a darker color for that node. The distance between nodes is irrelevant, it was adjusted accordingly to depict the graph. The information in a directed graph can be overwhelming to analyze for every simulation run, therefore, we use its information to calculate a macro-property reflecting their variability in phase space.







**Figure 1.** EC-Earth SST system state identification and evolution during the PI experiment. a) the three principal components (PCs) indexes series, blue for negative and red for positive, dashed lines indicating the 1 standard deviation value. From top to bottom, the AEM (first component), AMM (second Component), and SASD (third component). b) To the right, the cluster identification of the system state given the three above PC indexes. To the left, the cluster identification of the system state is represented by a directed graph. Each node signifies a specific state, as depicted in Figure S1 from Supplementary Material

*Entropy as an analogue for climate variability*

Shannon's Entropy (equation 1) is used in this study to measure the variability from each simulation in the phase space by attributing its trajectory with an entropy value.



## EC-Earth precipitation evolution in the PI experiment

**a)**



**b)**

**Figure 2.** EC-Earth precipitation system state identification and evolution during the PI experiment. a) the three principal components (PCs) indexes series, blue for negative and red for positive, dashed lines indicating the 1 standard deviation value. From top to bottom, the first, second, and third components. b) To the right, the cluster identification of the system state given the three above PC indexes. To the left, the cluster identification of the system state is represented by a directed graph. Each node signifies a specific state, as depicted in Figure S1 from Supplementary Material

$$H = -\sum_{k=1}^{N} P(x_k) ln(P(x_k)) \tag{1}$$

The entropy of a dynamic system is a measure of organization. For example, the systems from Fig. 3 'a' and 'b' have the
same initial and final states (0 and 4, respectively), however, system 'a' evolves directly from the initial to the final state, while



system 'b' varies across different states until arriving at the final state. This means that system 'a' is less chaotic, its trajectory is more organized, and hence its entropy is smaller.

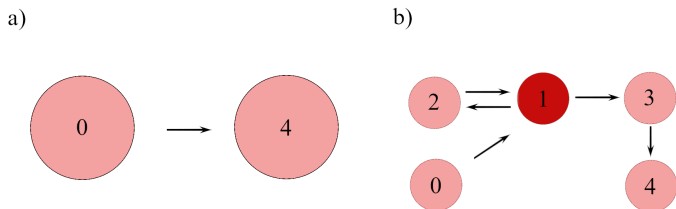

**Figure 3.** Two abstract directed graphs representing dynamic systems of the same time series length: a) A system evolves from its initial state (0) to its final state (4) — a low-entropy system. b) A system evolves from its initial state (0) to state 1, then to state 2, back to state 1, to state 3, and finally to its final state (4) — a high-entropy system.

Looking at equation 1, the probability ($P(x_k)$) of our system $x$ (the simulated Atlantic Ocean SST and precipitation) being found in each possible state ($k$) can be calculated empirically from the simulation time series. For example, if a specific simulation has been in only one state during its whole time series, that state ($x_k$) has a probability equal to one to be found in that specific state and zero on the others. Considering that $\sum_{k=1}^{N} P(x_k) = 1$, the entropy from that time series is the lowest possible, zero. Using the 27 states from the discrete PC phase space as the possible states of our system, the entropy from each time series will reflect its variability in that space. Since we are using the same space for all the models and scenarios, we can compare their variability.

From a broader perspective, the probability distribution from the Atlantic SST or precipitation patterns (micro-states) in a simulation (trajectory) is used to determine the systems' decadal variability (macro-properties).

## 2.3 Proposed approach for measuring uncertainty

There are a few sources of uncertainty when calculating entropy from numerical climate model outputs. Numerical models inherently involve: (i) internal variability - the general climate variability at pre-industrial or other standard scenarios -; (ii) differences in numerical discretization - which shape how physical processes are represented -; and (iii) scenario-based uncertainty (Lehner et al., 2020). Our analysis focuses on variability emerging from the discrete PC phase space representation. Since we use a unified phase space for all models and scenarios, our calculated entropy reflects differences in how each simulation explores this unified space. The uncertainty in entropy is derived empirically, based on how entropy depends on the PC space and the threshold values used. While we do not explicitly separate internal variability, discretization, and scenario uncertainty, our approach indirectly captures elements of all three. However, since we are not using a large ensemble, our conclusions are limited to the specific simulations analyzed.

In this study, we did not compare observational data to the model output patterns, but, in theory, we could. In this scope, the observational data variability and the different model scenarios variability could be compared using the same methodology.





Furthermore, the observation and model representations need to be balanced when defining a common PC phase space. This
space creates the base where the system will be projected, and later, the trajectories and entropy will be calculated. If biased
systems are compared, it is best to balance the data set used to derive the PCs, otherwise, instead of a trajectory's organization,
a low entropy may reflect the absence of representation of its variation in the PC phase space.

### *The Possible States of the Tropical and South Atlantic system*

The choice of representing our system with the 3-dimensional PC space was based on previous literature about how the modes
of variability work at decadal time scales. The decadal SST patterns known as modes of variability are correlated to the ocean
and continental precipitation such that these patterns' feedback and interaction with each other are the main drivers of the
Tropical and South Atlantic decadal variability (Gorenstein et al., 2023). By using this PC space we are discussing numerical
model variability at different experiments with the same metrics used to discuss observational data variability.

### *Entropy's Maximum and its Uncertainty*

Entropy is an emergent property of the 3-dimensional PC phase space. A common way to define the positive, negative, and
neutral phases of each PC index is to apply a fixed threshold, such as half a standard deviation. However, small changes in
this threshold can significantly alter the simulation's trajectory through phase space and thus affect its entropy (see Figures
S2 and S3 from Supplementary Material). Instead of using a fixed rule, we identify, for each simulation, the threshold that
yields the maximum entropy—i.e., the threshold that results in the greatest diversity of system states during the simulation.
This "maximum entropy" reflects the most representative variability for that simulation. Because this ideal threshold varies
between simulations, we use each simulation's maximum entropy as a consistent basis for comparing variability. We estimate
the uncertainty of this value using a bootstrap approach.

In each simulation, an interval of ideal thresholds was applied to define the 27 possible states according to the intensity of
the three PCs. The maximum entropy value was attributed to each time series, and a bootstrap technique was used to define
its 95% confidence interval (the percentile bootstrap interval) (Hinkley, 1988; Diciccio and Romano, 1988; Gorenstein et al.,
2022b). To calculate this confidence interval, we augmented our data using only our pre-existing dataset for each experiment,
and, with this augmented data set, 1000 PC indexes were generated, from which we calculated their maximum entropy and
their 95% confidence interval. This confidence interval was taken as the uncertainty from each trajectory's entropy and used to
discuss the climate variability of each simulation run.





## 3 Results

We define our system as the Tropical and South Atlantic decadal Sea Surface Temperature (SST) and precipitation. We analyze monthly outputs from 17 simulation runs across different experiments, each spanning 1200 months on a one-degree resolution grid. Therefore, each snapshot of the Tropical and South Atlantic has latitude and longitude dimensions of 80x89. To focus on the large climate patterns in our interest region and reduce the dimensionality in our data, we derive the three main Principal Components (PC) and use them to define two phase spaces, one for SST and another for precipitation. The PC analysis identifies the main patterns of our system's variability (Haykin, 2009). Furthermore, we project our system in these phase spaces, creating an index measuring the temporal evolution of each PC component during the simulation ('a' from Figs. 1 and 2).

When looking at the SST, the first three PCs are also known as the modes of variability of the Tropical and South Atlantic. The Atlantic Meridional Mode (AMM), the Atlantic Equatorial Mode (AEM), and the South Atlantic Subtropical Dipole (SASD). These modes' indexes represent our system in a low-dimensional phase space. To transform this space (the AMM, AEM, and SASD indexes) into a discrete representation, we divide the PC indexes into three phases (positive, negative, and neutral), defining the 27 possible states in which our system's SST can be found (Figure S1 from Supplementary Material). We expect precipitation and SST variabilities to be highly coupled at decadal time scale (Gorenstein et al., 2023). Therefore, analogous to the SST possible states, a construction is made using the first three PCs of precipitation. For SST and precipitation, the system state evolution throughout each simulation can be represented by a trajectory in phase space.

In Figs. 1 and 2, the EC-Earth $PI$ experiment is used to illustrate how the PC indexes correlate with its trajectory. However, this choice is not particularly significant. All other graphs (Figs. 4, 5, 6, and 7) follow a similar construction and yield comparable results.

### 3.1 Directed Graphs

As mentioned before, each simulation run can be represented as two trajectories in the PC phase spaces, one for the SST and one for the precipitation ('b' from Figs. 1 and 2). These trajectories of the Tropical and South Atlantic systems are represented as directed graphs in each simulation experiment (Figs. 4 - 7). Each node in these graphs represents a system state at a given time step, with its size showing the duration spent in that state and its color intensity reflecting the frequency of transitions to and from other states (higher degree). Darker nodes indicate states the system revisits more often.

The evolution in time of the EC-Earth PI in the SST and precipitation highlights the Atlantic Ocean decadal SST and precipitation cycles around a main pattern (cluster 0 - the PCs neutral pattern - Figure S1 from Supplementary Material). The same properties used to design each graph from Figs. 4 - 7 were employed to calculate macroproperties such as the entropy of the time series.





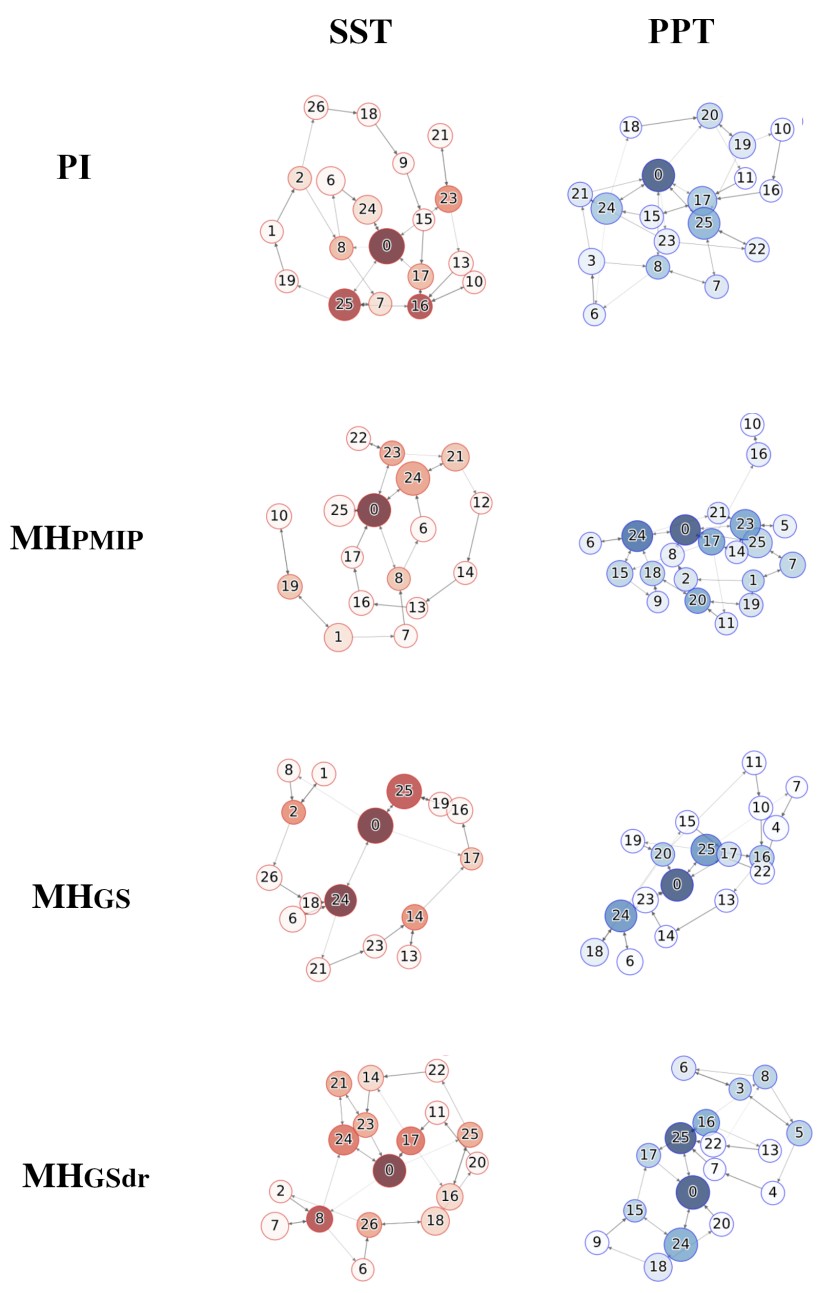

**Figure 4.** Directed graphs from EC-Earth - PI, $MH_{PMIP}$, $MH_{GS}$, and GS with dust reduction ($MH_{GSdr}$) simulations. The red graphs represent the SST system evolution, and the blue graphs represent the precipitation evolution. Each node signifies a specific state, as depicted in Figure S1 from Supplementary Material.





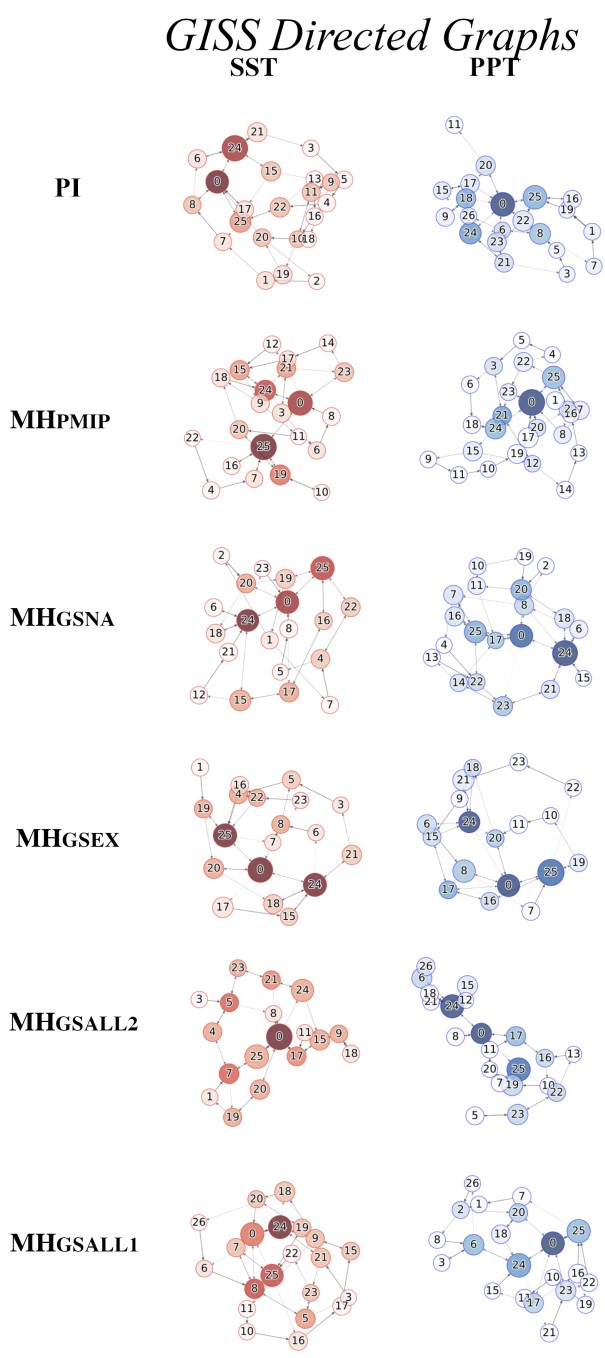

**Figure 5.** Directed graphs from GISS - PI, $MH_{PMIP}$, $MH_{GS}$ with North Africa ($MH_{GSNA}$), Extra-Tropical ($MH_{GSEX}$), and full vegetation ($MH_{GSALL1}$ and $MH_{GSALL2}$) runs. The red graphs represent the SST system evolution, and the blue graphs represent the precipitation evolution. Each node signifies a specific state, as depicted in Figure S1 from Supplementary Material.





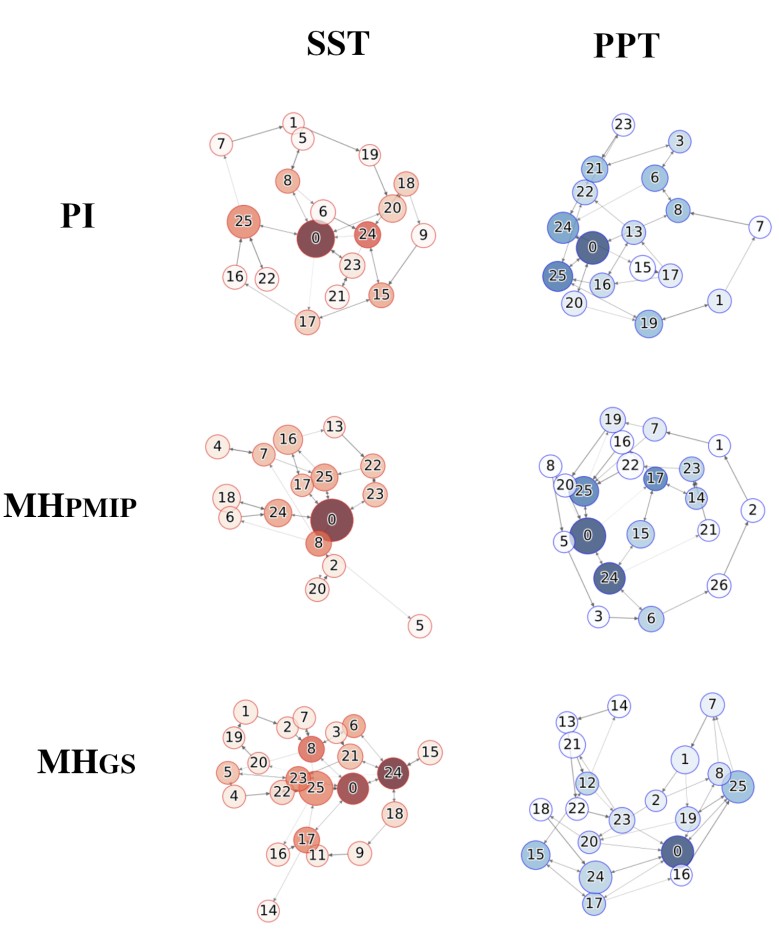

**Figure 6.** Directed graphs from iCESM - PI, $MH_{PMIP}$, and $MH_{GS}$ runs. The red graphs represent the SST system evolution, and the blue graphs represent the precipitation evolution. Each node signifies a specific state, as depicted in Figure S1 from Supplementary Material.



# CCSM-Toronto Directed Graphs

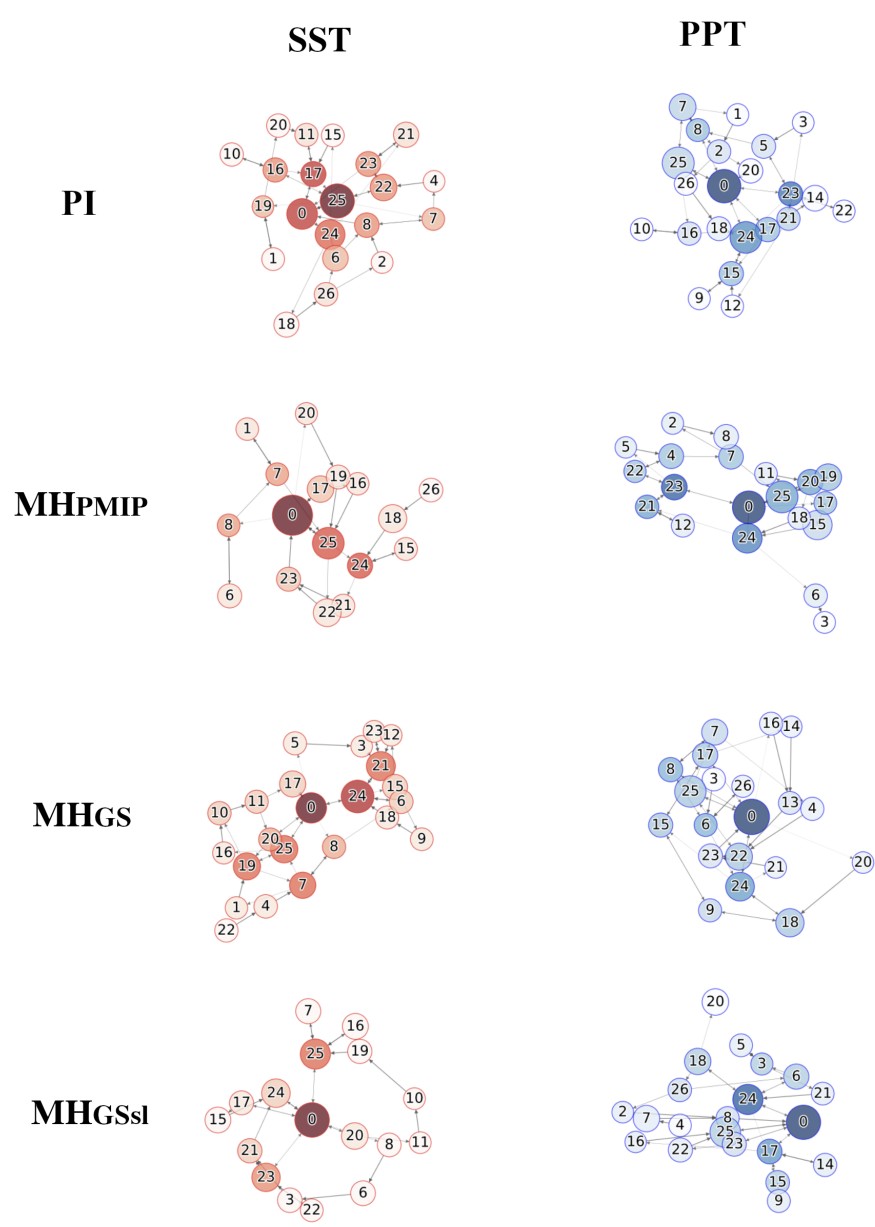

**Figure 7.** Directed graphs from CCSM-T - PI, $MH_{PMIP}$, $MH_{GS}$, and $MH_{GSsl}$ with soil and lake input runs. The red graphs represent the SST system evolution, and the blue graphs represent the precipitation evolution. Each node signifies a specific state, as depicted in Figure S1 from Supplementary Material.





## 3.2 Shannon's Entropy and Model Variability Analysis

In this study, Shannon's Entropy ($H_{sst}$ and $H_{ppt}$) is used to assess the level of organization within the tropical and South Atlantic system based on its SST and precipitation PCs (as described in Equation 1).

Our choice to study the decadal variability comes from the SST and precipitation coupling at this time scale. Since these two variables are correlated and have a strong feedback interaction controlling the energy balance along the Equator (Schneider et al., 2014), we expect them to vary together, creating a distinguishable climate variability response to external forces. From

230 a broader perspective, the entropy calculation in the PC phase space reflects the organization of the SST modes of variability and their precipitation counterparts in each simulation. From this point onward, we characterize climate variability in terms of the entropy of a simulation, reflecting the transition between different states. Table 2 shows the entropy and its 95% confidence interval for each model experiment.

Table 2: Entropy mean and Standard Deviation from the model runs.

| Model | Scenario | $H_{sst}$ | $H_{ppt}$ |
|-------|----------|-----------|-----------|
| ECE | $PI$ | $2.92 \pm 0.08$ | $3.00 \pm 0.06$ |
| ECE | $MH_{PMIP}$ | $2.98 \pm 0.04$ | $3.09 \pm 0.05$ |
| ECE | $MH_{GS}$ | $3.09 \pm 0.06$ | $2.96 \pm 0.07$ |
| ECE | $MH_{GSdr}$ | $2.94 \pm 0.06$ | $3.17 \pm 0.05$ |
| GISS | $PI$ | $3.06 \pm 0.06$ | $3.15 \pm 0.05$ |
| GISS | $MH_{PMIP}$ | $3.14 \pm 0.05$ | $3.16 \pm 0.05$ |
| GISS | $MH_{GSall1}$ | $3.20 \pm 0.05$ | $3.10 \pm 0.07$ |
| GISS | $MH_{GSall2}$ | $3.10 \pm 0.06$ | $3.04 \pm 0.05$ |
| GISS | $MH_{GSex}$ | $3.02 \pm 0.05$ | $3.17 \pm 0.06$ |
| GISS | $MH_{GSna}$ | $3.19 \pm 0.04$ | $3.12 \pm 0.05$ |
| iCESM | $PI$ | $2.98 \pm 0.06$ | $3.06 \pm 0.06$ |
| iCESM | $MH_{PMIP}$ | $3.15 \pm 0.06$ | $3.18 \pm 0.05$ |
| iCESM | $MH_{GS}$ | $3.17 \pm 0.05$ | $2.92 \pm 0.07$ |
| CCSM-T | $PI$ | $3.11 \pm 0.05$ | $3.17 \pm 0.05$ |
| CCSM-T | $MH_{PMIP}$ | $3.09 \pm 0.06$ | $3.03 \pm 0.05$ |
| CCSM-T | $MH_{GS}$ | $3.05 \pm 0.06$ | $3.01 \pm 0.06$ |
| CCSM-T | $MH_{GSsl}$ | $3.01 \pm 0.07$ | $3.05 \pm 0.06$ |

Despite differences in parameterizations and physics among models, each entropy was computed in a unified PC phase space (the states of every simulation were computed using the same EOFs). All model experiments were simulated over the same duration (100 years), enabling comparisons across the different models.

### *EC-Earth*

The EC-Earth model exhibits the lowest SST variability in the $MH_{PI}$ run (Fig. 8 'a'), indicating that in this scenario the

240 EC-Earth model simulates a more organized Atlantic SST system. The SST and precipitation variability from this model



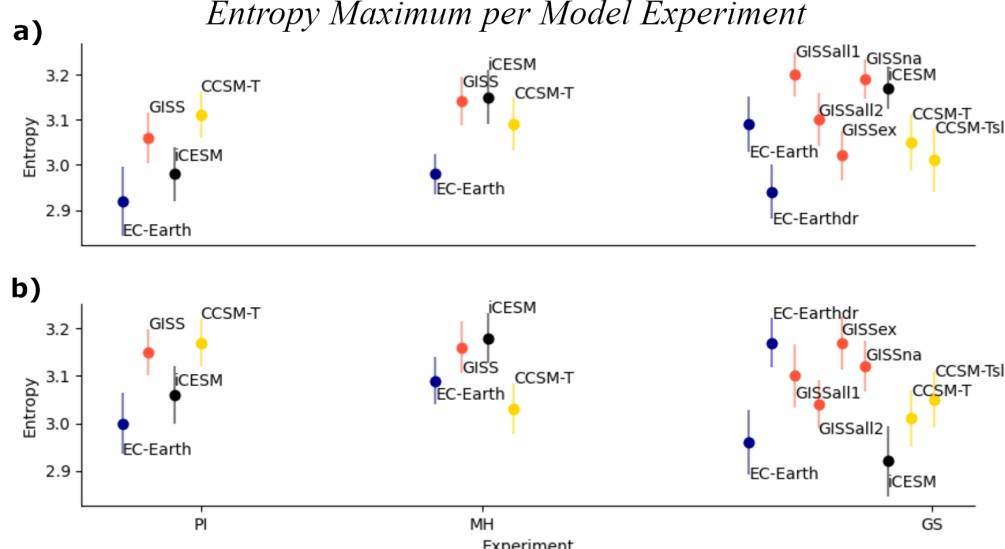

**Figure 8.** Entropy mean values and uncertainty for each model experiment. Entropy values for the SST (a) and precipitation (b) for Pre-Industrial ($PI$); mid-Holocene only orbital forcing ($MH_{PMIP}$); and mid-Holocene with Green Sahara boundary conditions ($MH_{GS}$) experiments. All models include PI, $MH_{PMIP}$, and $MH_{GS}$ runs. Under Green Sahara conditions additional experiments include: EC-Earth with northern African vegetation and dust reduction (dr); GISS with both extratropical and northern African vegetation (GISSall1 and GISSall2), with only extratropical vegetation (GISSex), and with only north African vegetation (GISSna); CCSM-T with soil and lakes (sl).

show opposite responses to the Green Sahara vegetation when dust reduction is considered. In the Green Sahara experiment without dust reduction ($MH_{GS}$), the precipitation variability is low, and comparable to the PI run, while the SST variability is significantly higher (5% in comparison to the $PI$). In the experiment where both vegetation and dust reduction are considered ($MH_{GSdr}$), precipitation variability shows an increase (6% in comparison to the $PI$ and 8% in comparison to $MH_{GS}$), while the SST variability decreases to $PI$ standards.

*GISS*

Compared to the $PI$ experiment, GISS exhibits significant changes in the SST variability only in the $MH_{GSall1}$ and $MH_{GSna}$ experiments (5% in comparison to the $PI$). While the precipitation variability has shown a decrease in $MH_{GSall2}$ (4% in comparison to the $PI$).



### *iCESM*

For this model, the lowest decadal SST variability happens in the $PI$ experiment. The $MH_{PMIP}$ and $MH_{GS}$ show no significant difference between them, they both have $6\%$ higher than $PI$ decadal SST variability. However, the precipitation variability is higher in the $MH_{PIMIP}$ experiment ($4\%$ higher than the $PI$ experiment and $9\%$ higher than the $MH_{GS}$).

### *CCSM-Toronto*

The decadal SST variability is higher in the $PI$ experiment, but not significantly different from the other experiments. Conversely, the $MH_{PMIP}$, $MH_{GS}$ and $MH_{GSsl}$ present $5\%$ lower than $PI$ precipitation variability.



## 4 Discussion

The standard approach to analyze SST and precipitation variability in climate reconstructions using proxies and numerical model simulations is to compute the two-dimensional standard deviation fields and frequency spectra (Flato et al., 2014; Olonscheck and Notz, 2017; Pendergrass et al., 2017). This approach accounts for the local dependencies of climate variables, where the standard deviation fields indicate the amplitude of regional variability, and the frequency spectra reveal periodicity.

Utilizing various types of biochemical proxy data from sediment and ice cores, previous studies have examined mid-Holocene climate variability in the Tropical and South Atlantic regions using the conventional methodology (Debret et al., 2009; Wirtz et al., 2010). Wirtz et al., (2010) found that most of the tropical Atlantic exhibited lower precipitation variability than present, except along the Northeast Brazil coast, where variability was higher. Additionally, numerical climate models have been used to recreate ocean modes indexes and study monsoon changes in Africa, America, and South America with mid-Holocene forcing parameters (Harrison et al., 2003; Zhao et al., 2007). An analysis of nine coupled ocean-atmosphere general circulation model experiments from PMIP2 by Zhao et al. (2007) indicated lower precipitation variability in the Sahel region during the mid-Holocene, with a weakening of teleconnections between Pacific/Atlantic SST and Tropical Atlantic precipitation, suggesting a decoupling of precipitation and SST.

Some of the model responses in our study agree with a lower precipitation variability in the tropical and South Atlantic during the mid-Holocene (specifically for CCSM-Toronto and iCESM's $MH_{GS}$) and suggest a decoupling between SST and precipitation variability when mid-Holocene vegetation in the Green Sahara is factored in (as seen with EC-Earth). However, there is no trivial path for comparison between the standard methodology and ours, since the concepts used to define climate variability differ.

The standard approach is more closely related to data-science statistics, effectively identifying regions most vulnerable to variability changes, which facilitates model regional validation with observational data. In contrast, our methodology measures a region's decadal climate variability concerning the ocean modes of variability and their precipitation counterparts, aligning more closely with the classical concepts of oceanography, climatology, and dynamic systems theory (Deser et al., 2010; Ghil and Lucarini, 2020). Modes such as El Niño, the AMM, and the AEM influence climate across the globe, they are known to impact society (McGowan et al., 2012; Lam et al., 2019), agriculture (Anderson et al., 2018), the atmosphere (Xie and Carton, 2004; Gorenstein et al., 2023), and climate equilibrium (Pillai et al., 2022; Cai et al., 2021). Their evolution is a more conceptual framework to measure, discuss, and compare numerical models' decadal climate representation. Although we employed standard deviation to define the positive, neutral, and negative phases of the Atlantic modes, it is not necessarily the case that a simulation with high regional standard deviation (the traditional measure of climate variability) will correspond to a high entropy measurement. A simulation with high entropy exhibits oscillations between the states that define the discrete PC phase space.



# 5 Conclusions

Using the PC analysis to reduce the dimensionality of the data, we examine the decadal variability of Tropical and South Atlantic SST and precipitation across four different numerical models: EC-Earth, GISS, iCESM, and CESM-Toronto. We depict each simulation's SST and precipitation trajectories within a reduced PC phase space. While directed graphs illustrating the trajectories are intriguing, they can be overly complex. Therefore, it is more effective to use them to calculate macro properties, such as Shannon's Entropy, which serve as measures of system organization and provide analogs for the decadal variability observed across these simulations.

The highest entropy among the $PI$ runs is observed in the CCSM-Toronto and GISS models, attributed to the system evolving into a greater number of states within a shorter time frame. Moreover, the difference between the four models in $PI$ was not maintained across the different experiments, indicating a lack of consensus on the increase or decrease of variability in any specific experiment.

The CCSM-Toronto showed lower than $PI$ precipitation variability in the $MH_{PMIP}$ and $MH_{GS}$ runs, but no significant changes in the SST variability amongst the different experiments. The EC-Earth climate variability exhibited high sensitivity to dust emissions and vegetation in the Green Sahara. Comparing $MH_{PIMIP}$ and $MH_{GS}$, the vegetated simulation increased the SST variability while reducing the precipitation variability. This behavior inverts when comparing the $MH_{GS}$ and $MH_{GSdr}$ runs: the dust reduction parametrization lowered the SST variability while increasing significantly the precipitation variability. The GISS model displayed the smallest precipitation entropy variation among all experiments, showing no significant differences. The SST variability shows significant differences between $MH_{ex}$ and $MH_{na}$. Suggesting that the Atlantic SST variability increases with the presence of vegetation in North Africa, while the extra-tropical vegetation contributes to a decrease in SST variability. iCESM SST variability showed higher than $PI$ variability in the $MH_{PMIP}$ and $MH_{GS}$ runs, and the lowest entropy value of all models and experiments in its precipitation entropy from the $MH_{GS}$ experiment. When comparing the $MH_{PMIP}$ and $MH_{GS}$, although the SST variability remained similar, the precipitation variability presented a significant decrease (9%, the largest entropy variation from simulations within the same model). Indicating that the model's Tropical and South Atlantic precipitation variability has a high sensitivity to vegetation parametrization, however not directly affecting its SST variability.

To define the decadal variability of a simulation, we used the probability distribution from the ocean modes and their precipitation counterparts. We employed this technique for model comparison, furthermore, our methodology can be used to study model biases and validation. Accurately simulating large patterns and climate oscillations around its mean state can be challenging for numerical models, the methodology presented in this study adds a valuable tool to the comprehensive analysis of climate models.



*Code and data availability.* The current version of the models used to produce the results used in this paper are available from: EC-Earth - Döscher et al. (2021); iCESM - Tabor et al. (2020); CCSM-Toronto - Peltier and Vettoretti (2014); GISS - Schmidt et al. (2014). The exact
simulation outputs used to produce the results used in this paper can be found in the following link https://github.com/IuriGorenstein/Entropy_MH_ESM, as are scripts to produce the plots for all the simulations presented in this paper (Gorenstein, 2025).

*Author contributions.* All authors have made substantial contributions to this manuscript related to their areas of expertise.

*Competing interests.* The authors declare that none of the authors has any competing interests.

*Acknowledgements.* This study was financed in part by the Coordenação de Aperfeiçoamento de Pessoal de Nível Superior - Brasil (CAPES)
- Finance Code 001; FAPESP 2019/08247-1.

The authors would like to thank Dr. Qiong Zhang for providing the EC-Earth mid-Holocene experiments, as well as the constructive comments and suggestions that improved this manuscript.



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
