# Peer review of "The Atlantic Ocean's Decadal Variability in mid-Holocene Simulations using Shannon's Entropy"

_EGUsphere, 2025_

## Referee Comment (RC2)

**Review**

Title: **The Atlantic Ocean's Decadal Variability in mid-Holocene Simulations using Shannon's Entropy**

Author(s): Iuri Gorenstein, Ilana Wainer, Francesco S. R. Pausata, Luciana F. Prado, Pedro L. S. Dias, Allegra N. LeGrande, Clay R. Tabor, and William R. Peltier

MS No.: egusphere-2025-921

MS type: Methods for assessment of models

Iteration: Minor revision

The presented study has many strengths. First and foremost, it introduces an innovative approach that is rarely seen in the literature and aligns with established methodologies for assessing climate variability. The use of Shannon entropy to analyze climate variability based on trajectories in phase space is a novel method that goes beyond traditional metrics. Constructing a phase space using the first three principal components (PCs) of SST and precipitation, derived from PCA, is consistent with physically justified modes of variability (AEM, AMM, SASD). The comparative value of the study is enhanced by the use of four different models (EC-Earth, GISS, iCESM, CCSM-Toronto) and multiple scenarios (PI, MHPMIP, MHGS, etc.). Analyzing SST and precipitation separately enables the identification of potential decoupling in their response to different forcings. The application of a bootstrap approach to estimate confidence intervals for entropy is methodologically sound.

However, the study is not without flaws. One contradiction lies in the implicit assumption that high entropy equates to high physical variability. Shannon entropy measures the diversity of system states, but not necessarily the amplitude of fluctuations. A simulation with low-amplitude variability but frequent state changes may yield high entropy, despite low physical variability.

Another notable shortcoming is the lack of validation against observational data, even though the authors acknowledge that such a comparison would be possible. This is a critical point — without observational benchmarks, we cannot determine whether the models' entropy values are realistic or merely reflect internal simulation dynamics. A further difficulty is the inconsistency in model parametrization. The models differ in terms of the factors they include (e.g., vegetation, dust, lakes), making comparisons challenging. The study lacks an attempt to isolate partial effects — for example, what specifically causes changes in entropy: dust, vegetation cover, or their combination?

Moreover, the study does not quantitatively separate different sources of uncertainty. Although three types are mentioned — internal variability, discretization, and scenario-based uncertainty — their individual contributions to total variability are not assessed.

While the selection of three principal components may be reasonable, the study does not examine the sensitivity of results to the inclusion of additional components.

The use of maximum entropy for each simulation as a reference point is statistically understandable but may lead to non-comparable thresholds and obscure differences stemming from less dynamic models. This approach might favor models that "artificially" gain entropy through threshold adjustments.

Lastly, the graphical representation of results as directed graphs is visually complex and difficult to interpret.

One final comment, offered with all due respect and goodwill: a common PCA analysis should be performed for all models and for each variable (SST and precipitation) using a merged dataset from all models and experiments. This would ensure a shared phase space and resolve the issue of cross-model comparability.

2025-12-26